# Inpatient midwifery staffing levels and postpartum readmissions: a retrospective multicentre longitudinal study

Lesley Yvonne Turner ,[1] Christina Saville ,[1] Jane Ball ,[1] David Culliford,[1,2] Chiara Dall'Ora ,[1] Jeremy Jones,[1] Ellen Kitson-Reynolds ,[1] Paul Meredith,[1] Peter Griffiths [1,2]

[1]School of Health Sciences, University of Southampton, Southampton, UK
[2]NIHR Applied Research Collaboration Wessex, Southampton, UK

**Correspondence to**
Lesley Yvonne Turner;
lyt1g19@soton.ac.uk

## ABSTRACT

**Background** Preventing readmission to hospital after giving birth is a key priority, as rates have been rising along with associated costs. There are many contributing factors to readmission, and some are thought to be preventable. Nurse and midwife understaffing has been linked to deficits in care quality. This study explores the relationship between staffing levels and readmission rates in maternity settings.

**Methods** We conducted a retrospective longitudinal study using routinely collected individual patient data in three maternity services in England from 2015 to 2020. Data on admissions, discharges and case-mix were extracted from hospital administration systems. Staffing and workload were calculated in Hours Per Patient day per shift in the first two 12-hour shifts of the index (birth) admission. Postpartum readmissions and staffing exposures for all birthing admissions were entered into a hierarchical multivariable logistic regression model to estimate the odds of readmission when staffing was below the mean level for the maternity service.

**Results** 64 250 maternal admissions resulted in birth and 2903 mothers were readmitted within 30 days of discharge (4.5%). Absolute levels of staffing ranged between 2.3 and 4.1 individuals per midwife in the three services. Below average midwifery staffing was associated with higher rates of postpartum readmissions within 7 days of discharge (adjusted OR (aOR) 1.108, 95% CI 1.003 to 1.223). The effect was smaller and not statistically significant for readmissions within 30 days of discharge (aOR 1.080, 95% CI 0.994 to 1.174). Below average maternity assistant staffing was associated with lower rates of postpartum readmissions (7 days, aOR 0.957, 95% CI 0.867 to 1.057; 30 days aOR 0.965, 95% CI 0.887 to 1.049, both not statistically significant).

**Conclusion** We found evidence that lower than expected midwifery staffing levels is associated with more postpartum readmissions. The nature of the relationship requires further investigation including examining potential mediating factors and reasons for readmission in maternity populations.

## STRENGTHS AND LIMITATIONS OF THIS STUDY

⇒ Longitudinal design with individual level linking of staffing exposure to subsequent readmissions.
⇒ Adjustment for case mix, staff groups and organisational level variables using hierarchical multivariable modelling.
⇒ Staffing was measured in the first two 12-hour intervals of the index (birth) hospital stay and did not account for the exposure to staffing for the whole hospital stay.
⇒ Considers exposure to staffing in the whole maternity service rather than close to women at ward level.

complications is a key priority, not only because of the negative experience and burden on the women affected,[2] but because readmissions are costly.[3] Readmission rates are used as an indicator of the quality of care,[4] and in some countries, providers are held financially accountable for patient outcomes during the first 30 days after discharge for many patient groups.[5] Rates of postpartum readmission vary by country, with reported rates of 1.0% in the USA,[6] 1.7% in Sweden[7] and 3.3% in England and Wales.[8]

Reports from England and the USA suggest that readmission rates have increased in recent years.[8 9] This coincides with an increase in caesarean birth rates.[10] In England, 4.3% of women having a caesarean birth are readmitted (1 in 23) compared with 2.9% of those having a vaginal birth (1 in 34).[8] The recent trend towards shorter length of hospital stay and reduced community provision may also contribute to increased readmission rates. However, while some research has pointed to an association between reduced length of stay and increased readmission rates,[11 12] this finding is not consistent in the research literature.[13 14] The most common reasons for readmission include wound complications, postpartum haemorrhage, hypertension, mastitis and thromboembolism.[6 9 10 15 16]

## BACKGROUND

In many countries of the world, the majority of mothers give birth in hospital.[1] Preventing readmission due to potentially avoidable

Postpartum readmissions typically occur within the first 14–20 days after discharge.[17 18]

Pressures on staffing within maternity services have been reported globally,[19] but the consequences of this are poorly understood. In many areas of healthcare, there is strong evidence that lower nurse staffing levels are associated with reduced quality of care and poorer patient outcomes (eg, as measured by mortality,[20] adverse events[21] and patient satisfaction).[22] Studies in medical and surgical inpatient settings have found that higher registered nurse staffing levels are associated with reduced readmission rates.[23–25] A scoping review of midwifery and nurse staffing for inpatient maternity services[26] found just two studies which measured maternal readmission in relation to staffing. These studies found that readmission rates were lower when more midwifery staff were employed[27] and the proportion of registered staff increased.[28] More evidence is needed on the relationship between maternity staffing levels, skill mix and readmission rates to understand whether previous findings are replicated and to understand the size of the effect if an association exists.

The aim of this study is to assess the relationship between individual patient-level exposure to staffing and postpartum readmission rates. A longitudinal design was adopted to enable women's exposure to staffing to be aligned to their outcomes, adjusting for individual risk factors.

## METHODS

We conducted a retrospective longitudinal study using individual patient records in multiple centres.[29] Maternity inpatient stays from 13 April 2015 to 29 February 2020 were extracted from the hospital Patient Administration System for three Acute National Health Service (NHS) Trusts in England. Data after February 2020 were not studied due to service changes during the covid pandemic. NHS Trusts comprise one or more hospitals operating as a group under a combined management. Two of the Trusts had inpatient services all on one site (antenatal, postnatal and labour ward). The third Trust had two geographical sites, comprising two maternity hospitals.

The available data included all women admitted to maternity wards, including those admitted via Day Assessment triage unit, totalling 113 002 admissions. Patient data were extracted for the date and time of admissions and discharges per maternity patient episode during the study period. The patient pseudonyms and dates were used to identify which postpartum women had been readmitted to the same maternity service for any cause within 7 days and 30 days of discharge. The primary reason for readmission was extracted from the International Classification of Diseases, Tenth Revision code for the readmission episode. To account for variation in case mix and comorbidities, we used the age band on admission, mode of birth from the Healthcare Resource Group (HRG) procedure code,[30] and the Standardised Hospital Mortality Indicator

(SHMI) which calculates an age-specific risk score based on comorbidities for a given reason for admission ('diagnosis').[31 32] HRGs are standard groupings of clinically similar treatments which use comparable levels of healthcare resource. Admissions without a birth were excluded from the final data set as they would not be eligible to have a postpartum readmission.

Shift-level staffing data were obtained from Trust electronic roster systems which recorded all worked shifts and the grades of staff. This data comprised the time worked excluding breaks for the Registered Midwife (RM) group and Maternity Assistant (MA) staff group and was matched to 12 hour periods, either 07.00 to 19.00 or 1900 to 07.00 to determine the staff time available in each time period. Staffing was measured by Hours Per Patient Day (HPPD) and this variable was generated by dividing the total worked time for the Staff group by the total ward occupancy as detailed in the measure guidance document.[33] All admitted individuals were included in the occupancy measure because they all contribute to workload, with neonates accounting for 40% of the admitted population. The HPPD was calculated for RM and MA staff separately, and the staff groups were combined as a measure of overall staffing per 12-hour period per day. Skill mix was calculated per shift using the RM worked seconds as a proportion of the total worked seconds, which encompassed RMs and MAs. Registered nurses were included in the RM totals as there were very few (<0.1%) and were of a similar grade to midwives. Data were cleaned to remove shifts where patient occupancy is zero seconds, and to remove shifts with RM HPPD outliers (defined as RM HPPD<0.5 or RM HPPD>48).

Staffing for only the index (birth) admission prior to the postpartum readmission was recorded. For the analysis, staffing was noted in the first two 12-hour periods of care in the Index admission (one interval of day shift and one of night shift). This method avoids recording staffing exposure for increased length of stay for a minority of complex cases. Longer time periods will skew the analysis as there is more opportunity to be exposed to understaffing. We defined the expected staffing as the mean HPPD for the maternity service in each Trust, and this was calculated separately for both day and night shifts. Each of the two 12-hour intervals of observed staffing (HPPD) was divided by expected staffing for the same period (HPPD) and these ratios were averaged for the first two 12 hour intervals, with values below 1 indicating a reduction in staff compared with expected levels. The HPPD was converted into the number individuals cared for per midwife and individuals per MA by dividing 24 hours by the HPPD for each group. This allows service planners to see the potential differences of altering staffing levels in absolute terms using variables familiar to them. A further measure of ward activity and workload was calculated. The variable 'turnover' equalled the total number of admissions plus discharges per Trust per 12 hour interval. Transfers between wards were not included in this measure. Mean turnover was calculated per Trust

and variables were created to represent higher than expected turnover compared with the Trust mean, by day and night shifts.

Univariable analyses on the relationship between independent variables and readmissions were performed, nested in hospital Trust. Multilevel logistic regression was then undertaken with the primary outcomes of postpartum readmission within 7 days and 30 days of discharge, nested in hospital Trust. Readmissions within 7 days were included as this may be more sensitive to variation in care quality during index admissions,[34 35] and longer time intervals increase the chance that readmission may be due to outpatient management.[36] The null model was the starting point and variables were added first based on indicators in the literature to suggest a potential effect on readmission (age, mode of birth), then staffing variables of interest (exposure to RM and MA staffing below the mean in the main analysis) and then remaining variables of SHMI risk, skill mix and turnover were added in sequence and only retained in the full model if they improved model fit using Akaike information criterion and Bayesian information criterion values (see online supplemental file S1 for model fit using this forward selection process). After the model fitting, only age, mode of birth and SHMI risk remained in the full model along with the staffing variables. For ease of

interpretation, we repeated the full model using absolute values of staffing expressed as the number of individuals cared for by each midwife and MA. We performed a secondary analysis comparing low and high levels of staffing compared with near mean staffing, classified as 95%–105% of the mean, to explore non-linear relationships. As some subgroups may have higher support needs and thus be more vulnerable to any adverse effects of low staffing, we performed a secondary analysis to explore staffing effects according to mode of birth.

All analyses were conducted in Stata (Release V.16. College Station, TX: StataCorp LLC).

## Patient and public involvement

This study had service user and clinical staff consultation when selecting the outcome measures of interest, and a lay member contributed to the project steering group.

## RESULTS

There were 64 250 maternal admissions that led to a birth between mid-April 2015 and the end of February 2020. The Trusts differed in size with 5260, 45 518 and 13 472 admissions leading to birth in each of the Trusts, respectively. In the cohort, 58.1% of women had a spontaneous vaginal birth, 12.9% assisted instrumental birth, 12.3% planned caesarean birth and 16.7% emergency caesarean birth (table 1).

Two thousand nine hundred and three women who gave birth were readmitted postnatally within 30 days of discharge (4.5%). Of those readmitted within 30 days, 70% were readmitted within the first 7 days and 89% within 14 days. Rates of readmission differed by birth method: 3.7% of women having spontaneous vaginal birth were readmitted, and rates were 5.9% for planned caesarean birth, 6.3% for assisted birth and 6.8% for emergency caesarean birth, respectively. The postpartum readmission rates following birth varied by Trust and were 1.8%, 4.9% and 4.3%. The mean length of stay for postpartum readmissions was 45.4 hours (median 22.5 hours, IQR 3–52 hours). Infection/sepsis was the most common primary reason for readmission (677/2903, 23.3%). Postpartum haemorrhage accounted for 316/2903 readmissions (10.9%). Some readmissions appeared to be healthy mothers accompanying their newborn, although this represented only 125/2903 (4.3%) of maternal readmissions (online supplemental file S2).

Staffing varied by Trust for those in our study cohort, with an average of 2.3, 3.6 and 4.1 individuals per midwife in the three services (this equates to 11.7, 6.9 and 5.9 HPPD, respectively). More staff were available on day shifts compared with night shifts. The proportion of RM hours per Trust varies from 70.7%% to 77.4% between the Trusts, and almost all the registered staff are classified as midwives where this data are available (online supplemental files S3–S5).

For two of the Trusts, the mean overall staffing in HPPD has shown an increase during the 5 year study period,

| Table 1 | Characteristics of the study population and their exposure to staffing |
|---|---|
| **Age** | |
| <20 years | 2099 (3.27%) |
| 20–24 years | 9862 (15.35%) |
| 25–29 years | 18 362 (28.58%) |
| 30–34 years | 20 114 (31.31%) |
| 35–39 years | 11 016 (17.15%) |
| 40+ years | 2797 (4.35%) |
| **Mode of birth** | |
| Assisted birth | 7764 (12.88%) |
| Emergency caesarean birth | 10 041 (16.66%) |
| Spontaneous vaginal birth | 35 040 (58.14%) |
| Planned caesarean birth | 7425 (12.32%) |
| SHMI | Mean 0.0035 SD 0.0319 |
| Turnover (admissions plus discharges per day) | Mean 56.716 SD 25.397 |
| Registered midwife staffing exposure (HPPD) | Mean 6.640 SD 2.351 |
| Maternity assistant staffing exposure (HPPD) | Mean 2.146 SD 1.045 |
| Skill mix (registered midwives as proportion of total staff) | Mean 0.757 SD 0.039 |
| Readmitted within 30 days of discharge | 2903 (4.52%) |
| Readmitted within 7 days of discharge | 2043 (3.18%) |

HPPD, Hours Per Patient Day; SHMI, Standardised Hospital Mortality Indicator.

**Table 2** Univariable analyses of readmission at 7 days and at 30 days from discharge

| | Readmitted within 7 days | | Readmitted within 30 days | |
|---|---|---|---|---|
| | OR | 95% CI | OR | 95% CI |
| Age category | | | | |
| <20 years | 1.009 | 0.770 to 1.323 | 0.994 | 0.787 to 1.256 |
| 20–24 years | 1.000 | Reference category | 1.000 | Reference category |
| 25–29 years | 0.954 | 0.828 to 1.100 | 1.013 | 0.897 to 1.143 |
| 30–34 years | 1.040 | 0.906 to 1.195 | 1.095 | 0.973 to 1.233 |
| 35–39 years | 1.043 | 0.893 to 1.219 | 1.125 | 0.986 to 1.284 |
| 40+ years | 1.476 | 1.193 to 1.827 | 1.454 | 1.208 to 1.750 |
| Mode of birth | | | | |
| Assisted birth | 1.838 | 1.622 to 2.083 | 1.828 | 1.642 to 2.036 |
| Emergency caesarean birth | 1.888 | 1.685 to 2.116 | 2.041 | 1.854 to 2.247 |
| Spontaneous vaginal birth | 1.000 | Reference category | | Reference category |
| Planned caesarean birth | 1.500 | 1.310 to 1.717 | 1.706 | 1.526 to 1.908 |
| Registered midwife staffing<mean | 1.065 | 0.974 to 1.166 | 1.041 | 0.965 to 1.123 |
| Maternity assistant staffing<mean | 0.984 | 0.898 to 1.077 | 0.980 | 0.908 to 1.059 |
| Overall staffing<mean | 1.054 | 0.964 to 1.154 | 1.041 | 0.964 to 1.123 |
| SHMI risk | 1.554 | 0.601 to 4.021 | 1.546 | 0.671 to 3.560 |
| Turnover>mean | 0.972 | 0.889 to 1.062 | 0.966 | 0.896 to 1.042 |
| Skill mix<mean | 1.017 | 0.928 to 1.113 | 0.979 | 0.907 to 1.057 |

SHMI, Standardised Hospital Mortality Indicator.

whereas in the third Trust, there is a stable level of overall staffing but a slight decrease in RM staff and increase in MA staffing over this period (approx. 4% change in each group). There was more ward activity in terms of admissions and discharges on day shifts compared with night shifts (online supplemental files S6 and S7).

## Univariable analyses on covariates and postpartum readmission

Age and the mode of birth were statistically significant predictors of postpartum readmission in univariable analyses, with older mothers and those having assisted or operative birth at higher risk (see table 2 and online supplemental file S8). Higher SHMI risk was associated with higher risk of readmission, but this was not statistically significant. Women exposed to below average RM staffing and overall staffing had increased odds for readmission, but this was not statistically significant. Having more admissions and discharges than expected, and a more diluted skill mix than average was not associated with increased postpartum readmissions at 7 days and 30 days.

## Relationship between staffing and postpartum readmission in multivariable models

Model fit was improved from the null model using a forward selection procedure with the addition of age category, mode of birth, RM and MA staffing and SHMI risk in that order. It was not improved with the addition of turnover and skill mix variables and therefore these variables are not shown in the full model. The variance inflation

factors were below two for all variables which suggests that they are not collinear. Missing data are minimal for the variables studied (online supplemental file S1).

Lower than expected midwifery staffing was associated with an 11% higher odds of postpartum readmissions within 7 days of discharge (adjusted OR 1.108, 95% CI 1.003 to 1.223). The effect was smaller and not statistically significant for readmissions within 30 days of discharge (adjusted OR 1.080, 95% CI 0.994 to 1.175). Lower than expected MA staffing was associated with a lower odds of readmission at 7 days (OR 0.957, 95% CI 0.866 to 1.057) and at 30 days (OR 0.965, 95% CI 0.887 to 1.049). These relationships were not statistically significant in the multivariable model (see table 3 and online supplemental file S9).

In models using absolute (rather than relative) staffing levels, an increase of one individual (mother or baby) per midwife was associated with a 6% increase in the odds of readmission at 7 days, aOR 1.063 (95% CI 0.960 to 1.177). There was no association with MA staffing (aOR 0.998, 95% CI 0.978 to 1.018). Neither relationship was statistically significant (online supplemental file S10)

Staffing was examined in categories to capture higher than average staffing as well as lower than average levels for readmissions within 7 and 30 days (table 4, online supplemental file S11). When grouped as three categories, higher midwifery staffing (105% of mean or more) was associated with reduced odds of readmission compared with the category containing the mean. The category of lower than mean midwifery staffing (95% of mean or lower) was associated with increased odds of

**Table 3** 7 day and 30 day readmissions in the full multivariable models (staffing relative to the mean per service, online supplemental file S9)

| Variable | OR readmission 7 days | 95% CI | OR readmission 30 days | 95% CI |
|---|---|---|---|---|
| <20 years | 1.066 | 0.810 to 1.402 | 1.065 | 0.840 to 1.350 |
| 20–24 years | 1.000 | Reference category | 1.000 | Reference category |
| 25–29 years | 0.914 | 0.790 to 1.057 | 0.969 | 0.856 to 1.098 |
| 30–34 years | 0.983 | 0.853 to 1.133 | 1.024 | 0.906 to 1.156 |
| 35–39 years | 0.983 | 0.837 to 1.153 | 1.046 | 0.913 to 1.198 |
| 40+ years | 1.368 | 1.099 to 1.703 | 1.318 | 1.089 to 1.594 |
| Assisted birth | 1.873 | 1.648 to 2.128 | 1.850 | 1.657 to 2.065 |
| Emergency caesarean birth | 1.912 | 1.701 to 2.148 | 2.077 | 1.883 to 2.291 |
| Spontaneous vaginal birth | 1.000 | Reference category | 1.000 | Reference category |
| Planned caesarean birth | 1.518 | 1.321 to 1.745 | 1.707 | 1.522 to 1.915 |
| Exposed to staffing below mean registered midwives (HPPD) | 1.108 | 1.003 to 1.223 | 1.080 | 0.994 to 1.174 |
| Exposed to staffing below mean maternity assistants (HPPD) | 0.957 | 0.866 to 1.057 | 0.965 | 0.887 to 1.049 |
| SHMI risk | 0.687 | 0.108 to 4.360 | 0.8868 | 0.2421 to 3.2480 |

HPPD, Hours Per Patient Day; SHMI, Standardised Hospital Mortality Indicator.

readmission. None of the relationships in the categorical analyses were statistically significant.

In the secondary analysis by mode of birth, exposure to midwifery understaffing was associated with increased odds of readmission after emergency and planned caesarean births and assisted births. The estimated effects were stronger in these groups than in the whole cohort (OR 1.113–1.315) although the result was only statistically significant for the assisted birth group. For spontaneous vaginal births, there was no association between

**Table 4** Multivariable analysis with staffing in categories above and below mean levels for readmissions within 7 days and 30 days of discharge

| Variable | OR readmission 7 days | 95% CI | OR readmission 30 days | 95% CI |
|---|---|---|---|---|
| <20 years | 1.068 | 0.812 to 1.404 | 1.066 | 0.841 to 1.351 |
| 20–24 years | 1.000 | Reference category | 1.000 | Reference category |
| 25–29 years | 0.914 | 0.790 to 1.057 | 0.970 | 0.856 to 1.099 |
| 30–34 years | 0.984 | 0.854 to 1.134 | 1.025 | 0.908 to 1.158 |
| 35–39 years | 0.984 | 0.838 to 1.154 | 1.046 | 0.913 to 1.199 |
| 40+ years | 1.370 | 1.101 to 1.706 | 1.319 | 1.090 to 1.595 |
| Assisted birth | 1.873 | 1.648 to 2.127 | 1.850 | 1.657 to 2.065 |
| Emergency caesarean birth | 1.911 | 1.701 to 2.147 | 2.076 | 1.882 to 2.290 |
| Spontaneous vaginal birth | 1.000 | Reference category | 1.000 | Reference category |
| Planned caesarean birth | 1.518 | 1.321 to 1.744 | 1.705 | 1.520 to 1.913 |
| Midwifery staffing | | | | |
| Low, 95% of mean or less | 1.056 | 0.943 to 1.182 | 1.002 | 0.911 to 1.102 |
| Near to mean 95%–105% | 1.000 | Reference category | 1.000 | Reference category |
| High 105% of mean or more | 0.973 | 0.862 to 1.098 | 0.957 | 0.865 to 1.060 |
| Maternity assistant staffing | | | | |
| Low, 95% of mean or less | 0.924 | 0.816 to 1.046 | 0.968 | 0.871 to 1.076 |
| Near to mean 95%–105% | 1.000 | Reference category | 1.000 | Reference category |
| High 105% of mean or more | 0.935 | 0.825 to 1.059 | 0.968 | 0.870 to 1.077 |
| SHMI risk | 0.693 | 0.111 to 4.335 | 0.889 | 0.243 to 3.247 |

See online supplemental files S11 and S12.
SHMI, Standardised Hospital Mortality Indicator.

understaffing and readmissions (online supplemental file S13).

## DISCUSSION

In this longitudinal study, we analysed individual patient stays and concurrent staff rosters, and we found an association between midwifery staffing below the mean level and higher rates of postpartum readmission. This effect was stronger and statistically significant for readmissions within 7 days of discharge, with an increase in odds of 11% for understaffing compared with the organisation mean staffing levels. We estimate that if a midwife's workload is increased by one additional person, the odds of readmission within 7 days increases by 6.3%, although CIs are wide and cross the null effect in this analysis of absolute staffing data. The direction of findings is in line with that of Gerova,[27] who found that increased number of full-time-equivalent midwives in a service was associated with reduced postpartum readmissions. These findings are also seen in the nursing literature, in medical, surgical and paediatric settings.[23 37]

The mean 30-day readmission rate in this study was 4.5% which is higher than rates reported in USA[6] and Sweden[7] where reports are between 1% and 2%. This difference may reflect service variability such as community provision and the threshold for readmission.[38] The readmission rate following birth is much lower than with medical and surgical patients.[39] This is likely to be because obstetric patients are healthier and younger than other patient populations and because the index case is often undergoing a normal physiological process rather than management of a disease or condition.[5] The readmission median length of stay was 22.5 hours, which represents a sizeable provision within services each year when multiplied by the rate of readmission.

We found wide variation in the number of individuals cared for by each midwife, from 2.3 per midwife in one organisation to 4.1 per midwife in another. It is important to remember that this figure spans the inpatient maternity service where some women will receive one-to-one care in labour[40] and others will be onwards including neonates with fewer staff per person than in the birthing environment. One of the Trusts had a smaller service than the others, while also having higher rates of staffing and lower readmission rates. Smaller services require more staff to ensure that services are available and to manage unscheduled demand.[41 42] Smaller services may also be treating fewer complex cases which could account for lower readmission rates.[38] These variations were controlled for in our hierarchical analysis and also in the case-mix adjustment, so we can be confident our analysis has taken account of these factors.

Risk factors associated with maternal postpartum readmission include age, ethnicity, parity, body mass index and mode of birth.[10 15] Individual characteristics were noted to account for most of the interhospital variance in postpartum readmission rates in the study by Clapp et al[13] and we saw further evidence of this as age and mode of birth were statistically significant predictors of readmission in all of our analyses. Staffing is a modifiable factor that has also been associated

with rates of readmission.[27 28] Having more people to care for can affect the quality of care, as some aspects can be missed or incomplete as staff prioritise and ration their time.[44] Some activities such as medicines administration are commonly prioritised, whereas patient teaching and discharge preparation may be seen as less urgent, and therefore more likely to be incomplete or missed.[37] This has been found in a number of studies including those in Malaysia,[45] Australia,[46] USA[37] and South Korea,[47] although not all of these studies are in maternity populations. Our secondary analysis suggested that women with more complex modes of birth are particularly vulnerable to the effects of understaffing.

The mechanisms for an association between staffing and readmissions need further study. Weiss et al[48] found a sequential pathway from high registered nurse staffing, to improved discharge teaching quality and readiness, to decreased odds of readmission for medical–surgical patients. However, this type of study has not been replicated in the maternity population as far as we are aware. Proposed mechanisms include reduced staff time for recognising deterioration,[49] time for education,[37] coordination of care,[24] hand hygiene compliance,[4 50] discharge planning[44] and medicines optimisation.[17] We found that infection or sepsis was listed as the primary diagnosis in 23% of the postpartum readmissions. Although this was the most common identifiable reason for readmission, it was not as high as that reported in the literature, where infection has been implicated in 62% of all postpartum admissions[10] and 82% of readmissions following caesarean birth.[16] Nevertheless, this is still a sizeable figure so mediating pathways from staffing linked to infection and then readmission may be worthy of exploring.

We found a statistically significant association between midwifery staffing below the mean level and higher rates of postpartum readmissions within 7 days of discharge. We went on to look at staffing levels in categories below and above the mean levels. This categorical analysis of midwifery staffing found fewer readmissions when services were staffed above mean levels and more readmissions when staffed below mean levels, although these were not statistically significant. Service planners should consider setting staffing above mean levels if these findings are confirmed, as it is possible that mean levels in themselves represent suboptimal understaffing.

There was no clear relationship between MA staffing levels and readmission rates in our study. It may be that MAs perform fewer sensitive activities which could be related to the risk of readmission, although there is very little published evidence on assistants in maternity services to compare our findings to.

### Strengths and limitations

There is limited evidence to determine optimal staffing levels in maternity services. Our research adds to the body of evidence by highlighting the association between staffing levels of midwives and rates of postpartum readmissions. Our analysis has taken account of case mix and between-hospital differences when estimating effects, but unfortunately, we were unable to adjust for other important factors such as ethnicity, parity, deprivation and body mass index

from the available data. A strength of our approach is the longitudinal and granular design, so we have been able to measure the individual's exposure to staffing per shift, rather than estimating via monthly or annual data. Our approach has an advantage over sequential panel studies as the longitudinal design allows direct individual-level linking of staffing exposure to readmission despite the time lag between these variables.

The amount of missing data is minimal, and therefore we believe this does not materially affect the conclusions drawn. A limitation of our study is that we have been unable to extract information on the staffing close to each woman at the ward level, as two of the Trusts only had staffing data available as a whole service. We measured staffing in the first 24 hours of the admission including birth and will not have captured the hours before discharge for all women. Some of the protective mechanism of adequate staffing could be related to the period prior to discharge, with the handover of care and provision of discharge information. Also we have been unable to account for postdischarge care in the community, which is likely to have a bearing on readmission rates if women have differing opportunities to seek early intervention elsewhere.[15] A further limitation is that our study did not include the rosters of medical staff due to differences in roster configurations. If this staff group is included in future models, this may modify the estimates seen for midwives and MAs.

## Areas for future research

Any measurable reduction in hospital readmission rates would offer significant financial benefits to hospitals.[5] The cost of increasing staffing could be offset by the potential savings in the future. Cost-effectiveness studies are needed to investigate the consequences of different staffing configurations in relation to the quality of care and related costs and savings. In future research, the components of services that help to reduce readmissions could be explored so resources are applied in areas known to be effective. This may mean targeting increased attention towards at-risk populations that account for a high proportion of readmissions.[10 18] The provision of additional staff alone may not be fully effective in reducing readmissions unless they are focusing activity on areas of need or critical activities. A more thorough investigation of reasons for readmission would need to be included in this future work.

## CONCLUSION

Low midwifery staffing in maternity units is associated with increased rates of readmissions within 7 days after discharge, but there was no adverse effect from lower assistant staffing. This is an important finding especially as readmission rates have increased in some countries in recent years, they are costly to services and distressing for families. Limiting the number of individuals cared for per midwife may have benefits beyond the inpatient stay and result in cost savings if readmissions can be reduced.

**Acknowledgements** The views expressed are those of the author(s) and not necessarily those of the National Institute for Health Research, the Department of Health and Social Care, 'arms-length' bodies or other government departments. Frankie Lambert (FL) made contributions to the work that fall short of ICMJE authorship criteria but we wish her contribution to be recognised. Please note that, for the purpose of open access, the author has applied a Creative Commons attribution license (CC BY) to any Author Accepted Manuscript version arising from this submission.

**Contributors** Funding acquisition: PG, CDO, JB, DC, JJ, PM, CS. CS and PM cleaned and linked data from different sources LYT, PG, CDO, JB, DC, JJ, PM and CS developed the methodology. LYT conducted the investigation and analysis and wrote the original draft. LYT, PG, CDO, JB, DC, PM, CS and EK-R reviewed the results and contributed to interpretation. PG, EK-R and JB supervised this project. DC provided statistical advice. LT is the study guarantor. All authors reviewed the manuscript.

**Funding** National Institute for Health Research Health Services & Delivery Research programme, award 128056. NIHR Applied Research Collaboration Wessex, no award number.

**Competing interests** None declared.

**Patient and public involvement** Patients and/or the public were involved in the design, or conduct, or reporting or dissemination plans of this research. Refer to the Methods section for further details.

**Patient consent for publication** Not applicable.

**Ethics approval** This study involves human participants and was approved by ERGO 52957 and IRAS 128056. The study includes routine data submissions, and identifying variables were removed prior to the data being made available to researchers.

**Provenance and peer review** Not commissioned; externally peer reviewed.

**Data availability statement** No data are available. Unfortunately, due to the sensitive nature of the data and data sharing agreements with the providers, we are unable to freely share the source data, but we guarantee its authenticity and the rigour of methods used in the analysis.

**ORCID iDs**
Lesley Yvonne Turner http://orcid.org/0000-0003-1489-3471
Christina Saville http://orcid.org/0000-0001-7718-5689
Jane Ball http://orcid.org/0000-0002-8655-2994
Chiara Dall'Ora http://orcid.org/0000-0002-6858-3535
Ellen Kitson-Reynolds http://orcid.org/0000-0002-8099-883X
Peter Griffiths http://orcid.org/0000-0003-2439-2857

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
