## [Reviewer comments · BMJ Open]

ARTICLE DETAILS

TITLE (PROVISIONAL)	Inpatient midwifery staffing levels and postpartum readmissions – a retrospective multi-centre longitudinal study
AUTHORS	Turner, Lesley; Saville, Christina; Ball, Jane; Culliford, David; Dall'Ora, Chiara; Jones, Jeremy; Kitson-Reynolds, Ellen; Meredith, Paul; Griffiths, Peter

VERSION 1 – REVIEW

REVIEWER	Kaufman, Menolly Oregon Health & Science University
REVIEW RETURNED	05-Sep-2023

GENERAL COMMENTS	Abstract Can the authors clarify when the midwifery staffing was measured? Is it midwifery staffing during the birth admission? At the time of readmission or both? Introduction For clarity, the authors may consider condensing or removing a few paragraphs. Based on the research question, the authors may consider starting the introduction with the third paragraph (line 33 – “Pressures on staffing...”). Then follow with what is known about causes of readmissions, what is known about staffing, and what is missing in the literature. Methods Based on the description provided by the authors, they conducted a retrospective cohort study (using administrative records to observe an exposure (i.e., staffing levels) then following them forward through time (i.e., 30 days postpartum)). They may consider citing the specific study design. Can the authors explain what a day assessment unit is? Can the authors clarify if women in their study can readmit to other maternity services other than the one they delivered at? Can the authors clarify what other staff may be on maternity wards and why they are or are not included? US readers may ask about nurses and physicians on maternity wards.
--

	Can the authors describe all other variables, other than the exposure and outcome, that were included in the analyses? Can the authors add more details about the variables used to account for case mix differences? What are these indices? How do they account for comorbidities associated with birth outcomes and postpartum risk? Results The authors may consider starting the first paragraph with the number of maternal admissions for birth during their study period. It's unclear what admissions are for maternity but not for birth. A small comment, Table 1 is a Bivariate analysis (characteristics by readmissions). The reader would benefit from a reorganization of the results. First, it's useful for the reader if table 1 describes the population by the exposure status or, in this case, by Trust. Further, the readers may be interested in the specific components of the case mix scores, rather than just the overall score. It would logically follow to describe the various staffing indices. Then finally, report the associations between staffing and readmissions. A small comment in Table 2, 1.00 is typically referred to as the reference group, rather than 'base.' Why does Table 3 only examine 7-day readmissions and not 30-day? Discussion The authors should consider what other data limitations are presenting in their study. For instance, the study is missing a number of socio-demographic variables that are highly correlated with birth outcomes and readmissions. How do the readmission rates from these Trusts compare to the rest of the country? What is the external validity of your study? Can it be generalized to other countries?
--	---

REVIEWER	Goffinet, François INSERM, UMR S953, Epidemiological research on perinatal health and women's and children's health
REVIEW RETURNED	20-Sep-2023

GENERAL COMMENTS	It is a clear and well-written paper on an important topic. The association between staffing levels and consequences in terms of health is a question regularly asked by professionals but studies are rare because the results are often difficult to interpret. Rather than studying the association between staffing levels and maternal and childbirth complications, which is always difficult to interpret, the authors chose readmission as the outcome. The methods are well described and I find the way of measuring the expected staffing in each trust very convincing. I have few comments and am very favorable to the dissemination of these original results to help professionals discuss with hospital management to improve
---

	the quality of care. The discussion is particularly interesting on the different points raised by these results. 1 I understand the benefits of using exposure over the first 12 hours but this is an unmentioned point of discussion. The protective mechanism of good staffing could on the contrary be over the last 12 hours when the midwives or assistant organize the exit with the delivery of information and treatments. 2 I may not have fully understood how admissions work in these trusts. Many women come to the maternity ward and do not go directly to the labor room because they are not in active labor. Some will leave without having given birth, others will go to the labor room 12 to 24 hours later only. The first 12 hours begin upon admission but for women who do not go directly to the labor room they are probably less “exposed” to the staff. 3 I don’t know the information system used but does Healthcare Resource Group and the Standardized Hospital Mortality Risk are systems adapted to perinatal issues for adjustment for perinatal specific comorbidities? 4 Some abbreviations are not usual and even if they are explained at the beginning of the paper it is sometimes difficult to follow the words (RM, MA, SHMI) 5 I have difficulty understanding how many maternity wards are participating in the study because the authors use the word trust. We have the impression that there are perhaps two maternity wards in one of the trusts but it is not very clear to me. Is it necessary to give the results by trust, because for the obstetrician reader, giving the result by maternity ward would be clearer 6 page 10: mode of delivrey seems more appropriate to me than method of birth
--	---

REVIEWER	Siddiqui, FARAH University Hospitals of Leicester NHS Trust
REVIEW RETURNED	03-Dec-2023

GENERAL COMMENTS	Well written paper on an important aspect on care. Whether readmission rates are related to staffing. Methods longitudinal study of three maternity services in England over a 5 year period. Staffing and workload calculated per shift. However, I really struggled with the methodology. 1) The study did not control for the complexity of the patient for example patients who have had a csection for sepsis are high risk of readmission irrespective of workforce. 2)The study does not consider, the populations's deprivation scores, whether they need an interpreter, BMI or available support in the community. 3) Similarly, are multiparous individuals less likely to be readmitted than those having their first infant. 4) Further, the methodology included all inpatient admissions, as such, have the authors considered an inpatient being induced at a time with poor staffing, but good staffing during the intrapartum
--

	and postnatal period, surely the readmission cannot be based on the 24 hours of antenatal care. 5) Unclear whether the readmission only consider the PN readmissions? 6) Does the staffing levels relate to staffing in labour or on the postnatal ward. For the postnatal ward between 2.2 and 4.1 individuals per midwives sounds very light, but very extensive for intrapartum care where usually 1:1 midwifery is advocated. 7)Unclear regarding support for the postnatal infant impacts care, if the postnatal midwife is also looking after a ward full of transitional care infants, does this impact care and postnatal readmissions? The study may be better just studying postnatal patients and staffing levels.
--	--

REVIEWER	Matsumura, Kenta University of Toyama, Public Health
REVIEW RETURNED	22-Dec-2023

GENERAL COMMENTS	Statistical review comments  – Abstract: Describe staffing units (hours worked per patient day). – How many people did low (95% of mean or less), mean (95%-105% of mean), and high (105% of mean or more) staffing each consist of? Please report. Additionally, please explain why the authors did this when researchers usually divide into quartiles or triads, or based on SD. – Table 1: Please describe what was used as the reference category so that it is easy to understand just by looking at the table. For example, what is the reference for mode of delivery? Spontaneous vaginal delivery? So what has an OR = 0.849? Please describe this information in an easy-to-understand manner, such as in Table 2. – “age_category_combined”(Table 1 and S8.1 Table): Please state specifically how they were combined. Same for others. – The table should be independent of the text, and at a minimum, abbreviations should be placed in footnotes. – The description "inverted U-shape" is given, but since there is no significant difference among such a large number of people, it would be more appropriate to say that there is no difference at any level. Recommend deletion of the description "inverted U-shaped." – Since the tables in the supplement are pasted directly from the STATA output, they are not clear to all but the reader who does not use STATA. A processed and clean table should be created based on these values so that many readers can understand them. – Please attach a flowchart of the data selection (see the STROBE statement). – For important variables, please define them clearly and conspicuously to the reader (with units) by inserting formulas for their definitions. Currently, the inverse and categorization sections are quite messy and difficult to read. – In the statistical analysis section, please describe all analyses performed in a neat and orderly fashion, without any skipping, and follow that by describing the results in that order. Additional analyses, etc., should also be explained in the methods beforehand, rather than described suddenly in the results.
---

REVIEWER	Straub, Heather
-----------------	-----------------

	University of Colorado, Obstetrics and Gynecology
REVIEW RETURNED	29-Dec-2023

GENERAL COMMENTS	This is an important research question about staffing levels related to patient readmission on labor and delivery. The primary concerns are related to unclear methods and results that do not appear to control for some serious confounders (i.e mix of neonate vs maternal care and a lot of missing ICD-10 data on condition for readmission). 1- There is no sample size calculation or indication as to why the timeframe from April 13 2015 - Feb 29 2020 was used. For the outcomes which were not statistically significant, was this because the study was under powered? It would be helpful to the reader to know why this timeframe was chosen and the limitations of it. 2- The data presented is only Registered Midwives and Maternity Assistant Staff groupings. Do these hospitals have no other provider for obstetric care? (OB/GYN, Family Medicine, Nurse Practitioner, PA?). If there are other OB providers, their staffing levels could impact the outcomes. This could be better clarified in the methods. 3- On page 4, the calculation for "Hours per Patient Day (HPPD)" is outlined. It is unclear why the seconds worked by the staff group divided by the total ward occupancy in admitted person seconds was multiplied by 24. Firstly, is this a standard measure of appropriate staffing and if so, it should be referenced as such. The 24 is confusing, is it for the 24 hr in the day? if it is, why are the seconds being multiplied by hours and not the number of seconds in the day (86,400)? 3- The methods mention that 40% of the workload was neonates. While they do contribute to the obligations of a RM the demands of a neonate compared to a laboring or postpartum women would likely be less. It was unclear that this was specifically looked at or controlled for in their analysis. It would be helpful to stratify HPPD per adult patient vs neonate. 4- Results: The results mention the readmission rates for SVD vs planned cesarean delivery vs assisted birth were different, but it is unclear what these are percentages of- are the percentage of readmitted patients? are they percentages of total births? This could be better clarified. 5- Infection/sepsis is listed as the most common reason for readmission, however, when you look at the ICD-10 codes, if you look at the number of "Other" diagnostic codes, non-specific readmission etiologies, "blank" or "supervision of normal pregnancies" a rough estimate shows that ~860/2903 patients had these diagnoses (~29%). This could absolutely impact the frequency of readmission. Further investigation into the reasons for readmission including possible direct chart review would strengthen the argument that the readmissions are correlated to staffing concerns.
--

	6- Throughout the paper the term "case-mix adjustment" is used and it was difficult to see a definition of this term and how it was used in the paper. 7- The supplementary material references different "Bands" (i.e 2/3/6) If this is included in the paper it should be defined. 8- The reason why readmission at 7 and 30 days should be towards the beginning of the methods section rather in its current location. 9- Are staffing ratio definitions of "low", "high" and "mean" standardized in the literature or in the approach to hospital staffing? If so, they should be referenced as such. If not, the reasons why these cutoffs were used should be explained.
--	---

VERSION 1 – AUTHOR RESPONSE

Point by point response to reviewers

Reviewer: 1	
Dr. Menolly Kaufman, Oregon Health & Science University, Portland, USA	
Abstract 1.1 Can the authors clarify when the midwifery staffing was measured? Is it midwifery staffing during the birth admission? At the time of readmission or both?	Thank you for this query. The midwifery staffing was measured during only the index admission (the admission which resulted in the birth). We did not examine staffing at the time of the readmission as the readmission itself was the endpoint and outcome of interest. We have made this clearer in the manuscript Lines 41, 75, 163 and S13
Introduction 1.2 For clarity, the authors may consider condensing or removing a few paragraphs. Based on the research question, the authors may consider starting the introduction with the third paragraph (line 33 – “Pressures on staffing...”). Then follow with what is known about causes of readmissions, what is known about staffing, and what is missing in the literature.	Thank you for your thoughts on improving the clarity and conciseness of the introduction. We have removed the text in lines 98-103 to make the introduction slimmer and easier to navigate. We have not reordered the paragraphs but feel the introduction reads better with some text removed.
Methods 1.3 Based on the description provided by the authors, they conducted a retrospective cohort study (using administrative records to observe an exposure (i.e., staffing levels) then following them forward through time (i.e., 30 days postpartum). They may consider citing the specific study design.	Thank you for highlighting this. Line 124-5 is the first line of the methods and states ‘We conducted a retrospective longitudinal study using individual patient records’. We have now included a citation for the protocol as this outlines the methods in more detail Line 125.
1.4 Can the authors explain what a day assessment unit is?	The day assessment unit is a triage and assessment service which usually manages outpatient cases

	but women can be admitted as inpatients via this service. We have added the word triage to Line 133
1.5 Can the authors clarify if women in their study can readmit to other maternity services other than the one they delivered at?	Thank you for this query. Yes, this is possible although it is rare for women to be readmitted to another NHS Trust. Readmissions to other Trusts would not be captured by this dataset. We have not discussed this as we estimate the numbers would be minimal and would not affect the overall findings. The methods section on Lines 134-136 says 'The patient pseudonyms and dates were used to identify which postpartum women had been readmitted to the same maternity service for any cause within 7 days and 30 days of discharge.'
1.6 Can the authors clarify what other staff may be on maternity wards and why they are or are not included? US readers may ask about nurses and physicians on maternity wards. (see 5.3 for similar point)	Thank you for highlighting this. We were unable to include medical staff (physicians) as they are not covered by the same rosters that we were able to access for this study. In England these doctors cover both obstetric and gynaecology specialties and there is some difficulty in assessing their obstetric contribution alone. We have added Lines 378-382 to the limitations section 'A further limitation is that our study did not include the rosters of medical staff due to differences in roster configurations. If this staff group is included in future models, this may modify the estimates seen for midwives and maternity assistants.' Nurses are grouped with midwives, but they are very few in total in England (explained in Line 159-160).
1.7 Can the authors describe all other variables, other than the exposure and outcome, that were included in the analyses?	Thank you for raising this. We have now added Lines 198-199 to the Methods to explain which variables were retained in the full model. 'After the model fitting, only age, mode of birth and SHMI risk remained in the full model along with the staffing variables'. This is also explained in the results section Lines 260-261 'Model fit was improved from the null model using a forward selection procedure with the addition of age category, mode of birth, RM and MA

	staffing and SHMI risk in that order. It was not improved with the addition of turnover and skill mix variables and therefore these variables are not shown in the full model.'
1.8 Can the authors add more details about the variables used to account for case mix differences? What are these indices? How do they account for comorbidities associated with birth outcomes and postpartum risk? (see 2.3 and 5.8 as similar point)	Thank you for raising this. We have amended Lines 137-141 to say 'To account for variation in case mix and comorbidities we used the age band on admission, mode of birth from the Healthcare Resource Group (HRG) procedure code³⁰, and the Standardised Hospital Mortality Indicator (SHMI) which calculates an age specific risk score based on comorbidities for a given reason for admission ('diagnosis').^{31,32} The references that are included will be useful to readers wanting to look at these measures in more depth.
Results 1.9 The authors may consider starting the first paragraph with the number of maternal admissions for birth during their study period. It's unclear what admissions are for maternity but not for birth. (see 3.4 and 4.8 as similar point)	Thank you, we have removed 'There were 113,002 separate maternal admissions' as this relates to all admissions, and is not as relevant to the study question and potentially confusing. Lines 214-215 Our study is based on the 64,250 maternal admissions that led to a birth and looked at the staffing they were exposed to, and the numbers readmitted postnatally. Supplement S13 has also been added to clarify the study population as the request of another reviewer.
1.10 A small comment, Table 1 is a Bivariate analysis (characteristics by readmissions).	Thank you for raising this. A commonly used term in logistic regression analysis is the term 'univariable' to describe the unadjusted association between a single variable and the outcome (Zhang 2016). We are aware that the term Bivariate is also sometimes used but have not chosen this terminology for the journal audience. Our statistical reviewer (reviewer 4) has not mentioned this point and therefore we think it may be possible to leave the text as originally written. Zhang, Zhongheng. "Model building strategy for logistic regression: purposeful selection." Annals of translational medicine 4.6 (2016).
1.11 The reader would benefit from a reorganization of the results. First, it's useful for the reader if table 1 describes the population by the exposure status or, in this case, by Trust. Further, the readers may be interested in the specific components of the case mix scores, rather than just the	Thank you for your suggestions and highlighting this oversight. We have made a new Table 1 to report on the characteristics of the population and staffing exposures for

overall score. It would logically follow to describe the various staffing indices. Then finally, report the associations between staffing and readmissions.	the whole study population. Details per Trust can be found in the supplementary material. Thank you, this now follows on in Tables 2-4 which have been renumbered from the original draft.
1.12 A small comment in Table 2, 1.00 is typically referred to as the reference group, rather than 'base.'	Thank you, this has now been amended in all Tables of results and the word 'base' has been replaced as suggested.
1.13 Why does Table 3 only examine 7-day readmissions and not 30-day?	Thank you for highlighting this omission, we have now added the data for readmissions within 30 days to this Table, which has now been relabelled Table 4. The results for 30 day readmissions are similar for those at 7 days with regards to staffing categories.
Discussion 1.14 The authors should consider what other data limitations are presenting in their study. For instance, the study is missing a number of socio-demographic variables that are highly correlated with birth outcomes and readmissions. (see 3.1 as a similar point)	Thank you. We have now highlighted this issue in the limitations 'Our analysis has taken account of case mix and between-hospital differences when estimating effects, but unfortunately, we were unable to adjust for other important factors such as ethnicity, parity, deprivation and body mass index from the available data.' Lines 363-365
1.15 How do the readmission rates from these Trusts compare to the rest of the country? What is the external validity of your study? Can it be generalized to other countries?	Thank you for asking us to include this helpful information. We have added 'The mean 30-day readmission rate in this study was 4.5% which is higher than rates reported in USA⁶ and Sweden⁷ where reports are between 1-2%. This difference may reflect service variability such as community provision and the threshold for readmission.³⁸ Lines 305-307

Reviewer: 2	
Dr. François Goffinet, INSERM, UMR S953, Maternité Port Royal, Hôpital Cochin Saint-Vincent-de-Paul, Paris	
It is a clear and well-written paper on an important topic. The association between staffing levels and consequences in terms of health is a question regularly asked by professionals, but studies are rare because the results are often difficult to interpret. Rather than studying the association between staffing levels and maternal and childbirth complications, which is always difficult to interpret, the authors chose readmission as the outcome. The methods are well described, and I find	Thank you for your time in reviewing our manuscript. We agree that there are few studies in this area and are pleased that you recognise this paper as a valuable addition to the literature. We are pleased that you were able to follow the methods and you feel that

the way of measuring the expected staffing in each trust very convincing. I have few comments and am very favourable to the dissemination of these original results to help professionals discuss with hospital management to improve the quality of care. The discussion is particularly interesting on the different points raised by these results.	the measurement of staffing was appropriate. Thank you for your support, we also believe that these results are worthy of dissemination and will add to the debate about staffing and the quality of care.
2.1. I understand the benefits of using exposure over the first 12 hours, but this is an unmentioned point of discussion. The protective mechanism of good staffing could on the contrary be over the last 12 hours when the midwives or assistant organize the exit with the delivery of information and treatments.	Thank you for highlighting this. We measured staffing over the first 2 periods of 12 hours (24 hours in total) during the admission which included birth. We agree that for some women staffing in the hours leading to discharge will not be captured if they had a longer length of stay. We have added 'We measured staffing in the first 24 hours of the admission including birth and will not have captured the hours before discharge for all women. Some of the protective mechanism of adequate staffing could be related in the period prior to discharge, with the handover of care and provision of discharge information'. Lines 374-377
2.2 I may not have fully understood how admissions work in these trusts. Many women come to the maternity ward and do not go directly to the labor room because they are not in active labor. Some will leave without having given birth, others will go to the labor room 12 to 24 hours later only. The first 12 hours begin upon admission but for women who do not go directly to the labor room they are probably less "exposed" to the staff.	Thank you for raising this. Women who have antenatal admissions which do not result in a birth have been excluded from the study sample. In some areas of England, there is telephone triage which reduces the number of women presenting when they are not in established labour. The total number of admissions in the period was 113,002 however we have focussed on only 64,250 maternal admissions that led to a birth. Line 214 and new Supplement 13 flow chart
2.3 I don't know the information system used but are Healthcare Resource Group and the Standardized Hospital Mortality Risk systems adapted to perinatal issues for adjustment for perinatal specific comorbidities? (see 1.8 and 5.8 as similar point)	Thank you. We have added in more references to explain the HRG and SHMI measures and have amended Lines 137-141 to say 'To account for variation in case mix and comorbidities we used the age band on admission, mode of birth from the Healthcare Resource Group (HRG) procedure code³⁰, and the Standardised Hospital Mortality Indicator (SHMI) which calculates an age specific risk score based on

	comorbidities for a given reason for admission ('diagnosis').^{31,32} The SHMI is not tailored specifically for maternity, however there would be limitations for any chosen index. There are 260 Clinical Classification Systems included in the SHMI, and we have checked that these include those relevant to the maternity population (category 106) and we have referenced the source of this information.^{31,32} Reference 32 is a new reference to support this point.
2.4 Some abbreviations are not usual and even if they are explained at the beginning of the paper it is sometimes difficult to follow the words (RM, MA, SHMI)	Thank you, we have examined all of the tables and added footnotes to explain any abbreviations. We have kept abbreviations to a minimum in the text as we appreciate it can sometimes be difficult to follow.
2.5 I have difficulty understanding how many maternity wards are participating in the study because the authors use the word trust. We have the impression that there are perhaps two maternity wards in one of the trusts but it is not very clear to me. Is it necessary to give the results by trust, because for the obstetrician reader, giving the result by maternity ward would be clearer	Thank you for this observation. We have used the word Trust to mean organisation. Unfortunately, the data on staffing was only available at an organisational level for two of the three organisations and therefore we were unable to complete ward level analyses. To help with understanding we have added the following to the methods 'Two of the Trusts had inpatient services all on one site (antenatal, postnatal and labour ward). The third Trust had two geographical sites, comprising two maternity hospitals.' Lines 129-130
2.6 page 10: mode of delivery seems more appropriate to me than method of birth	Thank you, we have altered Lines 247 and 261 to read 'mode of birth'. We have avoided the word delivery as there is some resistance to that word in the UK and the word birth is more woman-centred and acceptable. We agree the word 'method' is not commonly used in this context.

Reviewer: 3	
Dr. Farah Siddiqui, University Hospitals of Leicester NHS Trust	
Well written paper on an important aspect on care. Whether readmission rates are related to staffing. Methods longitudinal study of three maternity services in England over a 5 year period. Staffing and workload calculated per shift.	Thank you for endorsing this paper and considering in an important area to have studied.
3.1 The study did not control for the complexity of the patient for example patients who have had a csection for sepsis are high risk of readmission irrespective of workforce. (see 1.8 and 2.3 as similar points about the indices used) (see 1.14 for consideration of limitations as similar points raised)	Thank you for highlighting this. We were able to adjust for mode of birth, age and SHMI risk which takes some account of underlying diagnosis and comorbidity. We have also added to our limitations section 'Our analysis has taken account of case mix and between-hospital differences when estimating effects, but unfortunately, we were unable to adjust for other important factors such as ethnicity, parity, deprivation and body mass index from the available data.' Lines 363-365 We recognise that there are factors missing from our analysis and this should be clearly noted so readers can understand the limitations of our work.
3.2 The study does not consider, the population's deprivation scores, whether they need an interpreter, BMI or available support in the community.	Thank you, we have added this to our limitations Lines 363-365 We had previously mentioned not accounting for community provision and recognise this is a limitation. Lines 377-378
3.3 Similarly, are multiparous individuals less likely to be readmitted than those having their first infant.	Thank you, we agree that parity is an important variable that we would have liked to have included. We did seek to do this but were unable to, so have recognised this as a limitation (as above)
3.4 Further, the methodology included all inpatient admissions, as such, have the authors considered an inpatient being induced at a time with poor staffing, but good staffing during the intrapartum and postnatal period, surely the readmission cannot be based on the 24 hours of antenatal care. (see also 1.9 as similar point)	Thank you for highlighting this. We previously reported the number for all inpatient admissions at the start of the Results section (113,002) but have now removed this text as our exposure to staffing and analysis is based only on the admissions leading to a birth Line 214. Supplement 13 also added to clarify the study population This means that all other antenatal admissions are not in the analysis as

	they wouldn't be able to have a postnatal readmission. If length of stay is much longer than 24 hours then the exposure period may capture an earlier part in the woman's journey (say a prolonged induction leading to a birth). We have added a section into the limitations which discusses the exposure window and how we have not always captured the hours before discharge which could also have implications for readmissions (Lines 374-377). This is a limitation of only using 24 hours but on balance we felt it was the best way to conduct this analysis as most women have 24 hours of data to contribute. We did consider alternative methodology of considering exposure to understaffing during the whole hospital stay but this also poses methodological problems as the longer the measurement period the more chance of being exposed to understaffing which could bias the analysis. We have noted this in Lines 163-167
3.5 Unclear whether the readmission only consider the PN readmissions?	Thank you. Yes, the readmissions are only counted for women who have previously had a birth at their last admission and were readmitted within 30 days. This is reflected in Lines 134-136 of the methods 'The patient pseudonyms and dates were used to identify which postpartum women had been readmitted to the same maternity service for any cause within 7 days and 30 days of discharge'
3.6 Does the staffing levels relate to staffing in labour or on the postnatal ward. For the postnatal ward between 2.2 and 4.1 individuals per midwives sounds very light, but very extensive for intrapartum care where usually 1:1 midwifery is advocated.	Thank you for querying this. The staffing exposure window is the first 24 hours for the birth admission so this may mean different things for different individuals. We recognise it is a crude measure and it will depend on the timing of birth within that admission. We did not have access to the time of birth, and we were unable to track women's movement through the service and map to rosters for each area. We have written in the limitations Lines 372-374 'A limitation of our study is that we have been unable to extract information on the staffing

	close to each woman at the ward level, as two of the Trusts only had staffing data available as a whole service.' As the measure is of organisational level staffing then we feel that the HPPD levels represent the clinical picture, as the numbers represent staff to person ratio across all inpatient settings.
3.7 Unclear regarding support for the postnatal infant impacts care, if the postnatal midwife is also looking after a ward full of transitional care infants, does this impact care and postnatal readmissions?	Thank you. We have included infants in the population to recognise that they are present in the environment and will have some workload attached to them. We have been unable to estimate the acuity of the neonates to gauge how much workload this represents. We felt it would complicate the paper by adding in a discussion about transitional care neonates and other factors that may impact on staff time and workload.
3.8 The study may be better just studying postnatal patients and staffing levels.	We agree and we had an initial aim to calculate postnatal ward-level staffing exposure and readmission rates. This was not possible as the electronic rosters did not show the ward location for staff in two of the three services. We are also aware that the rosters do not always reflect staff movement during a shift, as midwives are moved to work on labour wards at times of peak activity. Our previous research has looked at postnatal ward staffing in relation to the experience of care, and this is an area we have a continued interest in. This data was from another source but was monthly data and could not be linked to readmission rates in our dataset.

Reviewer: 4	
Dr. Kenta Matsumura, University of Toyama, Japan	
Statistical review comments	
4.1 Abstract: Describe staffing units (hours worked per patient day).	Thank you, we have now modified this sentence in the abstract to include this 'Staffing and workload were calculated in Hours Per Patient day per shift in the first two 12-hour shifts of the index (birth) admission. ' Lines 40-41
4.2 How many people did low (95% of mean or less), mean (95%-105% of mean), and high (105% of mean or more) staffing each consist of? Please report. Additionally, please explain why the authors did this when researchers usually divide into quartiles or tertiles, or based on SD. (see 5.11 as similar point)	Thank you. For registered midwives the staffing categories breakdown is as follows Low, 95% of mean or less 39.6% Near to mean 95%-105% 29.5% High, 105% of mean or more 30.9% For maternity assistants the staffing categories breakdown is as follows Low, 95% of mean or less 42.7% Near to mean 95%-105% 20.3% High, 105% of mean or more 37.1% These tables have been added into the supplementary information for readers (S12) We chose to explore the relationship in more detail by looking at staffing levels slightly over and slightly under mean levels. This was to help understand the impact of variation that we might anticipate in practice, and to understand the effect of these changes in staffing. This grouping allowed the mean (plus 5% either side of the mean) to be considered an 'expected' level of staffing allowing for small amount of variability. The criteria for 'low' and 'high' were set outside this expected level. This seemed to be more meaningful that splitting the data into tertiles or quantiles as it is more reflective of a real clinical scenario. It was felt that organisation

	managers would better understand our thresholds than having cut offs determined by standard deviations, which we are aware is another commonly used method for categorisation. We have moved/added some text to explain this in the methods 'We performed a secondary analysis comparing low and high levels of staffing compared to near mean staffing, classified as 95% to 105% of the mean, to explore non-linear relationships.' Lines 201-203
4.3 Table 1: Please describe what was used as the reference category so that it is easy to understand just by looking at the table. For example, what is the reference for mode of delivery? Spontaneous vaginal delivery? So what has an OR = 0.849? Please describe this information in an easy-to-understand manner, such as in Table 2.	Thank you for pointing this out. We have reformatted all the tables in the same way as Table 2 with the reference categories for age and mode of birth being highlighted. This will aid interpretation instead of simply presenting the coefficient for the analysis. Lines 256, 275, 291
4.4 "age_category_combined"(Table 1 and S8.1 Table): Please state specifically how they were combined. Same for others.	Thank you, we have removed the word 'combined' as this refers to data management of the original spreadsheet which had additional categories for those under 20 years (10-14 and 15-19) and extra categories for those aged over 40 years (40-44, 45-49, 50-54, 55-59). As these were small numbers, we combined these peripheral categories. This level of detail is probably not needed in the manuscript or supplementary data as it relates mainly to data processing, so we have removed the word 'combined' for this reason.
4.5 The table should be independent of the text, and at a minimum, abbreviations should be placed in footnotes.	Thank you, we have examined all of the tables and added footnotes to explain any abbreviations. We have kept abbreviations to a minimum in the paper.
4.6 The description "inverted U-shape" is given, but since there is no significant difference among such a large number of people, it would be more appropriate to say that there is no difference at any level. Recommend deletion of the description "inverted U-shaped."	Thank you, we have deleted this sentence 'An inverted U-shaped pattern was observed with the point estimates for maternity assistant staffing when grouped as three categories.' and kept the sentence 'None of the relationships in the categorical

	analyses were statistically significant.' Lines 286-287
4.7 Since the tables in the supplement are pasted directly from the STATA output, they are not clear to all but the reader who does not use STATA. A processed and clean table should be created based on these values so that many readers can understand them.	Thank you for this feedback, we have now removed all the STATA output from the supplementary information and replaced this with typed tables to help readers understand them. We agree this improves the paper for readers.
4.8 Please attach a flowchart of the data selection (see the STROBE statement).	Thank you for this suggestion. We have now created a flow chart for data selection and included this in Supplementary material S13
4.9 For important variables, please define them clearly and conspicuously to the reader (with units) by inserting formulas for their definitions. Currently, the inverse and categorization sections are quite messy and difficult to read.	Thank you, the inverse calculation is given in the methods Lines 173-176 'The HPPD was converted into the number individuals cared for per midwife and individuals per maternity assistant by dividing 24 hours by the HPPD for each group. This allows service planners to see the potential differences of altering staffing levels in absolute terms using variables familiar to them.' Categorisation is also given in the methods section 'We performed a secondary analysis comparing low and high levels of staffing compared to near mean staffing, classified as 95% to 105% of the mean, to explore non-linear relationships.' Lines 201-203 We have ensured that these sections are fully explained by adding text to Supplements S3 and S11
4.10 In the statistical analysis section, please describe all analyses performed in a neat and orderly fashion, without any skipping, and follow that by describing the results in that order. Additional analyses, etc., should also be explained in the methods beforehand, rather than described suddenly in the results.	Thank you, we have now checked the methods to ensure that all analyses are explained. We have included the text for the univariable analyses 'Univariable analyses on the relationship between independent variables and readmissions were performed, nested in hospital Trust. Lines 187-188 We now have reordered the methods to ensure that analyses are introduced in the same order that the results are presented. Lines 187-203

	Thank you for pointing out this oversight and we feel that the methods are now much improved. The order of analyses are: 1. Univariable2. Multivariable with a relative measure of staffing to organisation mean3. Multivariable with an absolute measure of staffing (individuals cared for by midwife / maternity assistant)4. Multivariable with staffing in 3 categories with 'near mean' as the reference group
--	--

Reviewer: 5	
Dr. Heather Straub, University of Colorado, USA	
This is an important research question about staffing levels related to patient readmission on labor and delivery. The primary concerns are related to unclear methods and results that do not appear to control for some serious confounders (i.e mix of neonate vs maternal care and a lot of missing ICD-10 data on condition for readmission).	Thank you very much for your time in reviewing this and for your interest in this area. We have included your suggestions as much as possible but were limited by the data that was available in fully answering this question.
5.1 There is no sample size calculation or indication as to why the timeframe from April 13 2015 - Feb 29 2020 was used.	Thank you for your observation. The time frame of data collection related to the availability of rosters in a similar format for use in the analysis. Before April 2015 the roster data would be more difficult to clean and match it to admissions due to the way it was recorded. We ended the data collection period at the end of February 2020 as the staffing levels and services were altered due to the covid pandemic. We have added this explanation into Lines 127-128 'Data after February 2020 was not studied due to service changes during the covid pandemic.'
5.2 For the outcomes which were not statistically significant, was this because the study was under powered? It would be helpful to the reader to know why this timeframe was chosen and the limitations of it.	The sample size would have affected the precision of our estimates as reflected in the width of confidence intervals. We did not perform power calculations prior to the sample collection. This is not always performed when conducting research on retrospective observational data and we did not start with a hypothesis to test, unlike in clinical trials. We favoured assessing data for quality and completeness when choosing our sample.
5.3 The data presented is only Registered Midwives and Maternity Assistant Staff groupings. Do these hospitals have no other provider for obstetric care? (OB/GYN, Family Medicine, Nurse Practitioner, PA?). If there are other OB providers, their staffing levels could impact the outcomes. This could be better clarified in the methods. (see 1.6 for similar point)	Thank you for highlighting this. We were unable to include medical staff (physicians) as they are not covered by the same rosters that we were able to access for this study. In England these doctors cover both obstetric and gynaecology specialties and there is some difficulty in assessing their obstetric contribution alone. We have added Lines 379-382 to the limitations section 'A further limitation is that our study did not include the rosters of medical staff due to differences in

	roster configurations. If this staff group is included in future models, this may modify the estimates seen for midwives and maternity assistants.' Nurses are grouped with midwives, but they are very few in total in England (explained in Line 159-160).
5.4 On page 4, the calculation for "Hours per Patient Day (HPPD)" is outlined. It is unclear why the seconds worked by the staff group divided by the total ward occupancy in admitted person seconds was multiplied by 24. Firstly, is this a standard measure of appropriate staffing and if so, it should be referenced as such. The 24 is confusing, is it for the 24 hr in the day? if it is, why are the seconds being multiplied by hours and not the number of seconds in the day (86,400)?	Thank you for asking us to explain this further. The reason the multiplier of 24 was used is that when care and occupancy were in seconds, they were both using the same units of measurement (seconds). The Care Hours Per Patient Day measure has these variables in different units (care is in hours and the occupancy is in days), so the adjustment is necessary in data processing. We have removed some of the detail from Line 153 as this is not necessary in the main paper although we are happy that our calculations are correct. We have now referenced the CHPPD calculation as it is outlined in the NHS Digital supporting material. The new sentence reads 'Staffing was measured by Hours Per Patient Day (HPPD) and this variable was generated by dividing the total worked time for the Staff group by the total ward occupancy as detailed in the measure guidance document.³³ Lines 151-153
5.5 The methods mention that 40% of the workload was neonates. While they do contribute to the obligations of a RM the demands of a neonate compared to a laboring or postpartum women would likely be less. It was unclear that this was specifically looked at or controlled for in their analysis. It would be helpful to stratify HPPD per adult patient vs neonate.	Thank you for your thoughts on this. We felt it was important not to miss the neonates as there may be different numbers in each unit depending on the services offered, and we didn't want them to be 'invisible' as they do make up some of the workload. The natural variation in the ratio of women to neonates is unlikely to be related to the associations we are examining. Differences between Trusts in their population make up will be captured by the hierarchical analysis.

	We have not attempted to consider the workload attached to individuals with each population and are aware that there will be variation in workload for the population across the spectrum of age and clinical need. For these reasons we have not stratified HPPD for women and neonates separately.
5.6 Results: The results mention the readmission rates for SVD vs planned cesarean delivery vs assisted birth were different, but it is unclear what these are percentages of- are the percentage of readmitted patients? are they percentages of total births? This could be better clarified.	Thank you for highlighting that this was not clear. These percentages are for each of the populations for each respective group. We have amended the text in Lines 225-228 to make this clearer. 'Rates of readmission differed by birth method: 3.7% of women having spontaneous vaginal birth were readmitted, and rates were 5.9% for planned cesarean birth, 6.3% for assisted birth and 6.8% for emergency cesarean birth respectively.'
5.7 Infection/sepsis is listed as the most common reason for readmission, however, when you look at the ICD-10 codes, if you look at the number of "Other" diagnostic codes, non-specific re-admission etiologies, "blank" or "supervision of normal pregnancies" a rough estimate shows that ~860/2903 patients had these diagnoses (~29%). This could absolutely impact the frequency of readmission. Further investigation into the reasons for readmission including possible direct chart review would strengthen the argument that the readmissions are correlated to staffing concerns.	Thank you for this observation. We agree that it is a shame that the reporting is incomplete and not more specific within the data. We have added the following sentence to the section on future research 'A more thorough investigation of reasons for readmission would need to be included in this future work.' Lines 392-393
5.8 Throughout the paper the term "case-mix adjustment" is used, and it was difficult to see a definition of this term and how it was used in the paper. (see 1.8 and 2.3 as similar point)	Thank you, we have this located in Lines 137-141 'To account for variation in case mix and comorbidities we used the age band on admission, mode of birth from the Healthcare Resource Group (HRG) procedure code³⁰, and the Standardised Hospital Mortality Indicator (SHMI) which calculates an age specific risk score based on comorbidities for a given reason for admission ('diagnosis').^{31,32} HRGs are standard groupings of clinically similar treatments which use comparable levels of healthcare resource.' We have added in additional detail to this section as requested by yourself and other reviewers.
5.9 The supplementary material references different "Bands" (i.e 2/3/6) If this is included in the paper it should be defined.	Thank you for raising this. This 'banding' refers to the pay grade of staff and indicates seniority. This

	information is not used in the paper but readers may be interested in this data which is why we have retained it in the supplementary material. We have added some more detail in the supplementary material to explain the meaning of 'bands' (S5)
5.10 The reason why readmission at 7 and 30 days should be towards the beginning of the methods section rather in its current location.	Thank you for your suggestion. We currently have this on Line 136-137 in the second paragraph of the methods 'The primary reason for readmission was extracted from the ICD-10 code for the readmission episode.' This is quite early in the paper. As it is not the main focus of the paper, we feel that it is sitting in the right place for readers.
5.11 Are staffing ratio definition of "low", "high" and "mean" standardized in the literature or in the approach to hospital staffing? If so, they should be referenced as such. If not, the reasons why these cutoffs were used should be explained. (see 4.2 for similar point)	Thank you for highlighting this. This grouping allowed the mean (plus 5% either side of the mean) to be considered as an 'expected' level of staffing, allowing for a small amount of variability. The criteria for 'low' and 'high' were set to fall outside this expected level. This is more meaningful that splitting the data into tertiles or quantiles as it is more reflective of a real clinical scenario. It was felt that organisation managers would better understand our thresholds than having cut offs determined by standard deviations, which we are aware is another commonly used method for categorisation. We do not have a reference for this grouping but have now included the rationale in the Supplementary material S11. We have moved/added some text to explain this in the methods 'We performed a secondary analysis comparing low and high levels of staffing compared to near mean staffing, classified as 95% to 105% of the mean, to explore non-linear relationships.' Lines 201-203 A further explanation of the methodology is given in Supplement S11.

VERSION 2 – REVIEW

REVIEWER	Siddiqui, FARAH University Hospitals of Leicester NHS Trust
REVIEW RETURNED	29-Feb-2024

GENERAL COMMENTS	It would be helpful in the elective csection cohort to see if staffing effected the readmission rate as this group has less variables. Also just as with age it will be helpful to have information on the effect of BMI and previous caesarean section as this has been shown to have an impact on complexity. The statement "We estimate that if a midwife's workload is increased by one additional person, the odds of readmission within 7 days increases by 6.3%" appears quite bold, I can not see the statistical justification for this.
---

REVIEWER	Matsumura, Kent University of Toyama, Public Health
REVIEW RETURNED	01-Feb-2024

GENERAL COMMENTS	The authors adequately addressed all the comments; thank you.
---

VERSION 2 – AUTHOR RESPONSE

Reviewer: 3

Dr. FARAH Siddiqui, University Hospitals of Leicester NHS Trust

Comments to the Author:

It would be helpful in the elective csection cohort to see if staffing effected the readmission rate as this group has less variables.

Also just as with age it will be helpful to have information on the effect of BMI and previous caesarean section as this has been shown to have an impact on complexity.

Thank you for suggesting this improvement to our paper. We have now included the results for the subgroups of mode of birth and justified this as a secondary analysis in the methods (clean copy lines 188-190). The results have been added to the supplementary material S13 and are also described on lines 283-287 of the clean copy document. We have also added a line to the discussion (lines 331-333).

We have conducted the analysis for age subgroups but as there are six subgroups there are small numbers in some groups and results are less clear. This subgroup analysis has not been included in the paper but is available on request.

We did not have data on BMI and previous caesarean section but agree this data would be useful to collect in future studies.

The statement "We estimate that if a midwife's workload is increased by one additional person, the odds of readmission within 7 days increases by 6.3%" appears quite bold, I can not see the statistical justification for this.

Thank you for enquiring about this. The data to support this is shown in detail in Supplement S10 (OR 1.063, 95% CI 0.960, 1.177)

The text in the discussion highlights that this is a non-significant result "We estimate that if a midwife's workload is increased by one additional person, the odds of readmission within 7 days increases by 6.3%, although confidence intervals are wide and cross the null effect in this analysis of absolute staffing data" (clean copy lines 294-296)

We have not amended the manuscript and hope you agree that the text above is sufficient as it is.